# MiR-146a in ALS: Contribution to Early Peripheral Nerve Degeneration and Relevance as Disease Biomarker

**DOI:** 10.3390/ijms24054610

**Published:** 2023-02-27

**Authors:** Eleonora Giagnorio, Claudia Malacarne, Paola Cavalcante, Letizia Scandiffio, Marco Cattaneo, Viviana Pensato, Cinzia Gellera, Nilo Riva, Angelo Quattrini, Eleonora Dalla Bella, Giuseppe Lauria, Renato Mantegazza, Silvia Bonanno, Stefania Marcuzzo

**Affiliations:** 1Neurology IV—Neuroimmunology and Neuromuscular Diseases Unit, Fondazione IRCCS Istituto Neurologico Carlo Besta, Via Celoria 11, 20133 Milan, Italy; 2Ph.D. Program in Neuroscience, University of Milano-Bicocca, 20900 Monza, Italy; 3Neuroalgology Unit, Fondazione IRCCS Istituto Neurologico Carlo Besta, Via Celoria 11, 20133 Milan, Italy; 4Unit of Medical Genetics and Neurogenetics, Fondazione IRCCS Istituto Neurologico Carlo Besta, Via Celoria 11, 20133 Milan, Italy; 5Experimental Neuropathology Unit, Institute of Experimental Neurology (INSPE), Division of Neuroscience, San Raffaele Scientific Institute, 20132 Milan, Italy; 6Department of Clinical Neurosciences, Fondazione IRCCS Istituto Neurologico Carlo Besta, Via Celoria 11, 20133 Milan, Italy; 7Department of Medical Biotechnology and Translational Medicine, University of Milan, Via Vanvitelli 32, 20133 Milan, Italy

**Keywords:** amyotrophic lateral sclerosis, microRNA-146a, axon degeneration, biomarker

## Abstract

Amyotrophic lateral sclerosis (ALS) is characterized by the progressive, irreversible loss of upper and lower motor neurons (UMNs, LMNs). MN axonal dysfunctions are emerging as relevant pathogenic events since the early ALS stages. However, the exact molecular mechanisms leading to MN axon degeneration in ALS still need to be clarified. MicroRNA (miRNA) dysregulation plays a critical role in the pathogenesis of neuromuscular diseases. These molecules represent promising biomarkers for these conditions since their expression in body fluids consistently reflects distinct pathophysiological states. Mir-146a has been reported to modulate the expression of the NFL gene, encoding the light chain of the neurofilament (NFL) protein, a recognized biomarker for ALS. Here, we analyzed miR-146a and *Nfl* expression in the sciatic nerve of G93A-SOD1 ALS mice during disease progression. The miRNA was also analyzed in the serum of affected mice and human patients, the last stratified relying on the predominant UMN or LMN clinical signs. We revealed a significant miR-146a increase and *Nfl* expression decrease in G93A-SOD1 peripheral nerve. In the serum of both ALS mice and human patients, the miRNA levels were reduced, discriminating UMN-predominant patients from the LMN ones. Our findings suggest a miR-146a contribution to peripheral axon impairment and its potential role as a diagnostic and prognostic biomarker for ALS.

## 1. Introduction

Amyotrophic lateral sclerosis (ALS) is a fatal neurodegenerative disease characterized by the loss of motor neurons (MNs) in the motor cortex, brainstem, and spinal cord [1]. The ALS clinical spectrum includes extremely heterogeneous and complex phenotypes distinguished by a varying involvement of upper (UMN) and lower (LMN) MNs, site of onset, and rate of progression [2]. Of note, growing evidence demonstrated that terminal MN axonal degeneration, along with neuromuscular junction denervation, are early events in ALS pathogenic cascade [3,4]. Functional and morphological analysis showed early damage of the blood–nerve barrier followed by acute axonal degeneration associated with macrophage response in motor nerve compartments [3]. Progressive axonal degeneration and motor nerve fiber loss were found to correlate with magnetic resonance (MR) imaging and neurophysiologic changes in the ALS rat model [4]. By diffusion-tensor MR imaging, our previous study in the G93A-SOD1 mouse ALS model showed a gradient of degeneration in spinal cord white and gray matter, starting early in the ventral white matter, likely due to a cascade of early pathological events, including axonal dysfunction [5]. In the same model, straightforward evidence showed an early impairment of the peripheral nervous system compared to the central nervous system, with the presence of adaptive and innate immune cell infiltration [6]. In human patients, the relationship between motor axonal dysfunction at the early disease stage and disease progression was explored by compound muscle action potential amplitudes (CMAP), providing evidence that early peripheral motor axon dysfunction can influence ALS progression [7].

ALS can occur in two different forms: sporadic (sALS) in ∼90% of individuals and familial (fALS) [8]. Different genes have been associated with fALS and/or sALS: C9orf72–SMCR8 complex subunit (*C9orf72*) is the gene most commonly linked to inherited ALS, followed by TAR DNA-binding protein 43 (*TARDBP*), superoxide dismutase 1 (*SOD1*), and FUS RNA-binding protein (*FUS*) [9,10]. These genes affect several cellular functions, including RNA metabolism [11]. However, the exact molecular mechanisms implicated in ALS are not fully elucidated, and biomarkers of disease progression are not available yet. The identification of molecular biomarkers associated with disease course represents an important medical need since they could significantly improve the diagnostic process with relevant prognostic implications, aiding in patients’ stratification into distinct phenotypes and, where appropriate, in enrolling patients with a specific molecular signature in clinical trials [12,13]. To our knowledge, only the neurofilament light chain (NFL) protein, encoded by the neurofilament light chain (*NFL*) gene, has been identified as a promising disease biomarker. Indeed, CSF and serum NFL levels were found to be elevated in ALS patients [14,15,16] due to NFL release by the axonal plasma membrane as a consequence of axonal damage or degeneration [16]. Notably, NFL in CSF can differentiate ALS subgroups, in particular, bulbar compared to spinal onset ALS patients, as well as patients with *SOD* compared to *C9orf72* mutations, with *C9orf72*-ALS patients being characterized by higher plasma levels of NFL than *SOD*-ALS patients [17]. In a more recent study, the NFL levels were higher in patients with UMN predominance [14]; however, the exact molecular events underlying this increase in relationship with axonal degeneration are not entirely clear.

Growing evidence suggests that RNA metabolism alterations are critical for ALS pathogenesis [18,19]. MicroRNAs (miRNAs) are small non-coding RNAs that are key determinants of mRNA stability [20]. Several studies indicated a pivotal role for miRNAs in all aspects of neuronal development, function, and plasticity [21,22]. Moreover, the contribution of miRNA dysregulation to MN diseases, including ALS, is increasingly emerging [23,24,25,26,27]. Among miRNAs, miR-146a is known to negatively regulate the expression of the *NFL* gene [28], thus suggesting its potential involvement in ALS via *NFL* modulation. In particular, the role of miR-146a in the selective decrease in *NFL* mRNA and the formation of neurofilamentous aggregates in ALS has been described [28]. Furthermore, the elevated expression of miR-146a has been suggested to play a role in the inhibition of the production of inflammatory cytokines by antagonization of NF-κB activation and in the reduction of muscle mass, thus representing a molecular link between muscle atrophy and inflammation [29,30]. On the other hand, miR-146a deficiency has been associated with astrocyte and microglial cell transformation into the neurotoxic and proinflammatory phenotypes, which might contribute to MN degeneration, thus supporting the idea that miR-146a overexpression may exert a protective effect in ALS [31]. However, since miR-146a up-regulation in ALS patients’ spinal cords was associated with decreased NFL proteins that contribute to the maintenance of neuronal morphology [28], a side effect of miR-146a overexpression can be postulated. Nevertheless, whether miR-146a participates in early peripheral nervous system impairment, and specifically, in NFL dysfunction and axon degeneration at the peripheral nervous system in ALS, still needs to be elucidated.

Herein, we analyzed the expression of miR-146a in sciatic nerve and serum of G93A-SOD1 ALS mice at different disease stages by molecular analyses and in situ hybridization method. Moreover, to corroborate the potential role of this molecule as a non-invasive biomarker, we analyzed its expression in serum samples of ALS patients with signs of predominant axonal damage in UMNs and LMNs. We found an up-regulation of miR-146a in the sciatic nerve of affected mice that negatively correlated with transcriptional levels of the *Nfl* target gene, suggesting an association between the miRNA dysregulation and selective decrease in NFL in the sciatic nerve that could contribute to early axon degeneration in the peripheral nervous system. Contrariwise, the miRNA expression was down-regulated in the serum of affected mice, as well as in serum samples of human patients compared to healthy controls. These data suggested the involvement of miR-146a in human ALS, and its possible role, to be further explored as a therapeutic target to modulate the disease progression. Of note, serum miR-146a down-regulation was more prominent in patients with UMN than LMN predominant impairment, and the miRNA expression was significantly different between the two groups of patients, thus suggesting miR-146a potential value as a non-invasive biomarker to stratify ALS patients according to the disease phenotype.

## 2. Results

### 2.1. MiR-146a Increases and Nfl Gene Expression Decreases in Sciatic Nerve of G93A-SOD1 Mice during Disease Progression

To verify whether miR-146a expression was altered in the sciatic nerve tissue of G93A-SOD1 mice, we performed a longitudinal analysis of its expression levels by real-time PCR in control B6.SJL and affected G93A-SOD1 animals at the following disease stages: pre-symptomatic (week 8), onset (week 12), and symptomatic phase (week 18). Interestingly, our data revealed a significant up-regulation of miR-146a in G93A-SOD1 mice compared to controls (*p* < 0.05) already at week 8 (Figure 1a). MiRNA-146a levels were higher also at week 12, and particularly at week 18 (*p* < 0.05), in affected than control mice (Figure 1a). These results suggested that miR-146a overexpression in the sciatic nerve may be an early phenomenon in ALS, appearing before disease symptoms, then becoming evident and persisting during the disease course. 

To investigate possible alterations in *Nfl* expression in the sciatic nerve tissue of ALS animals, along with those of miR-146a, we performed real-time PCR analysis. We found that *Nfl* mRNA levels were significantly down-regulated in G93A-SOD1 compared to control animals at weeks 8 and 18 (Figure 1b, *p* < 0.05). These levels showed a trend to be also decreased at week 12 in affected versus control mice (Figure 1b), although differences did not reach statistical significance. Using Spearman’s correlation analysis, we assessed the relationship between miR-146a and *Nfl* expression in the sciatic nerve tissue of G93A-SOD1 mice. We found a significant negative correlation between the miR-146a levels and those of Nfl mRNA at the different time points (Figure 1c), suggesting a link between miR-146a increase in ALS sciatic nerve and *Nfl* reduction, which could contribute to axon degeneration.

To investigate the relationship between miR-146a-5p and the *Nfl* gene, we performed functional studies in NSC-34 motor neuron-like cells. By transfection with an miR-146a-5p inhibitor in NSC-34 motor neuron-like cells, we observed a significant up-regulation of the expression levels of the *Nfl* gene in miR-146a-5p inhibitor-transfected NSC-34 cells compared to scrambled-transfected cells (Figure 2, *p* < 0.05).

### 2.2. Increased miR-146a Levels in Sciatic Nerve of G93A-SOD1 Mice by In Situ Hybridization

We performed in situ hybridization to detect and localize miR-146a in the sciatic nerve tissue of G93A-SOD1 and control mice at weeks 8, 12, and 18. According to real-time PCR data, we showed a marked expression of miR-146a in the sciatic nerve of G93A-SOD1 compared to control mice already at week 8 and until week 18 (Figure 3a). The immunofluorescence intensity of U6 snRNA, examined as endogenous control, did not show a difference between the two groups of animals (Figure 3b). The scramble probe, a negative technical control, did not display any signals of fluorescence (Figure 3b). Increased expression of miR-146a in affected versus control animals was confirmed by quantification analysis of the miRNA immunofluorescence intensity normalized to that of U6 from weeks 8 to 18 (Figure 3c).

### 2.3. NFL Protein Decrease in Sciatic Nerve of G93A-SOD1 Mice

NFL protein levels showed a trend to be decreased in the sciatic nerve of G93A-SOD1 compared to control mice at weak 18, by Western blot analysis, as shown by quantification analysis of band intensity (Figure 4a,b, Appendix A). At weeks 8 and 12, we did not observe a decrease in the expression of NFL protein; these results are not in line with those of mRNA analysis, showing a decrease already at these weeks, which could be explained by the exceptional stability of NFL protein (approximately 3 weeks in mice optical axons) [32].

### 2.4. Decreased Expression of miR-146a in Serum Samples of G93A-SOD1 Animal Model and Human ALS Patients

To verify whether dysregulated expression of miR-146a in the sciatic nerve was accompanied by alterations of the circulating miRNA, we assessed its levels in the serum of G93A-SOD1 mice at the symptomatic disease stage, i.e., week 18, compared to control mice. We found that miR-146a levels were significantly down-regulated in sera of ALS compared to healthy mice (Figure 5), suggesting that the miRNA overexpression was a specific feature of the sciatic nerve and that it was associated with a reduction at the serum level.

Based on these findings, we wondered whether miR-146a levels may also be altered in ALS patients and whether the miRNA may represent a useful, non-invasive molecular biomarker in the disease. We thus assessed miR-146a expression in the serum of ALS patients and age- and sex-matched healthy donors (Table 1). 

As observed in affected animals, we found a significant down-regulation of miR-146a levels in ALS compared to control sera (Figure 6). This down-regulation was more evident in ALS patients with UMN compared to patients with LMN predominance, where the two groups did not differ in terms of disease duration at sampling (p = 0.797) (Figure 6a). By receiver operating characteristic (ROC) curve analyses, we obtained sensitivity and specificity diagnostic performance results, which supported a possible role for serum miR-146a as a disease biomarker for ALS, particularly in discriminating ALS patients with UMN from those with LMN predominance (Figure 6b). These findings suggest the potential role of miR-146a, to be validated in larger patients’ cohorts, as a potential biomarker for stratifying ALS phenotypes.

## 3. Discussion

The ALS clinical spectrum includes extremely heterogeneous phenotypes marked by a varying involvement of UMN and LMN, site of onset, and rate of progression [1]. The specific molecular determinants and potential biomarkers of disease progression remain to be fully elucidated. Thus far, only NFLs have been identified as promising disease biomarkers in ALS, reflecting axonal damage and degeneration [14,33]. Nevertheless, the exact molecular mechanisms linking NFL alterations and MN axon damage during the disease course are still unknown. Several data suggest that axonal degeneration already occurs at the early stages of ALS [34], supporting the need for a deeper understanding of the molecular changes implicated in early pathogenic events to develop efficient therapeutic approaches to counteract disease progression.

Most aspects of neuronal function and plasticity are regulated by miRNAs [22]. Among them, miR-146a is a crucial molecule for axon architecture and function [35]. Of note, this miRNA is known to modulate the expression of the *NFL* gene, which is a key gene for maintaining neuronal morphology in the spinal cord [28,36]. Based on this relevant function of miR-146a, in the present study, we analyzed its expression in the sciatic nerve of ALS mice during disease progression compared to control animals. Interestingly, we obtained real-time PCR data indicative of a significant up-regulation of miR-146a in the sciatic nerve of G93A-SOD1 ALS compared to control mice. This increase was already significant at the pre-symptomatic disease phase (week 8) and was maintained until the symptomatic phase (week 18). In situ hybridization performed in sciatic nerve sections of affected and control mice, confirmed a marked overexpression of the miRNA in the G93A-SOD1 mouse compared to the control group at early week 8 of life and throughout the disease course (week 12 and 18). We, therefore, suggested that early miR-146a dysregulation at the peripheral axonal level might be of relevance in the ALS pathogenic events, possibly contributing to the initial stages and perpetuation of axon degeneration. To deepen this hypothesis, we verified whether the miR-146a increase in ALS sciatic nerve was associated with dysregulated expression of genes implicated in axon function and structure. We focused on the *NFL* gene because it encodes for the light chain NFL protein, which is a crucial factor for the neuronal structure. Indeed, along with medium and heavy chains, the light chain is a component of NFLs, type IV intermediate filament heteropolymers that encompass the axoskeleton and functionally maintain the neuronal caliber, also playing a role in intracellular transport to axons and dendrites [37].

The ability to target the *Nfl* gene has been specifically demonstrated for miR-146-3p, known as miR-146a* [28]. Here, we showed a significant increase in miR-146a-5p in ALS mice, which could well be associated with decreased expression of *Nfl* since 5p/3p types of the same miRNA have been found to be co-expressed and co-target the same transcripts [38]. Moreover, the interaction between miRNA-5p and -3p with mRNAs have been shown to be fully complementary, as binding characteristics of different miRNA-5p/3p pairs in complementary binding sites of the same genes have been established [39]. Of note, *Nfl* mRNA levels showed an opposite trend than those of miR-146a, being significantly down-regulated in the sciatic nerve of ALS compared to control mice, already at week 8. In affected animals, reduced *Nfl* expression was maintained at week 12 and was marked at week 18. A significant negative correlation was found between miR-146a levels and the transcriptional levels of the *Nfl* gene, thus suggesting that decreased *Nfl* gene expression in the sciatic nerve of G93A-SOD1 mice might be related to abnormal miR-146a up-regulation at the early disease stage and during disease progression. Our functional studies in NSC-34 motor neuron-like cells by transfection with miR-146a inhibitor demonstrated a relationship between miR-146a-5p and Nfl gene expression, as observed for miR-146a-3p, suggesting that Nfl changes may be an effect of miR-146a-5p dysregulation in ALS mice. However, further functional studies are needed to clarify the exact relationship between miR-146a-5p and Nfl gene expression and whether this gene is a direct target of the -5p miRNA type, as observed for miR-146a-3p.

By biochemical analysis, we confirmed the lower expression of the NFL protein in the sciatic nerve of ALS mice at the symptomatic phase of the disease. These findings are in line with literature data showing that high levels of NFL protein in serum and/or CSF of patients with neurodegenerative diseases are not due to *NFL* overexpression (or over-production of its encoded protein) but to flaking and breaking of axons and subsequent release of NFL protein in biological fluids [40].

Our molecular data reveal miR-146a overexpression as an early molecular alteration occurring in ALS peripheral nerve, potentially implicated in axon impairment in association with reduction of *Nfl* gene expression and light chain NFL protein synthesis. Our findings open the hypothesis, to be deeply explored, that inhibiting the expression of miR-146a could directly or indirectly affect *Nfl* expression, potentially favoring motor axon re-organization and recovering neuronal integrity.

Since miRNAs are stable in body fluids and may reflect specific pathophysiological states or altered molecular mechanisms, they represent promising biomarkers [41,42]. These molecules can be released into the circulation by pathologically affected tissues and display remarkable stability in body fluids [43]. In the ALS field, the identification of miRNAs as disease biomarkers would be relevant to improve the diagnostic process and prognostic potential. Indeed, miRNAs could allow a more accurate diagnosis, giving the opportunity to start an earlier treatment with higher chances to modify the disease course. In addition, they could help in monitoring the disease progression and in the stratification of ALS patients in distinct pathological phenotypes to drive proper patients’ enrollment in clinical trials and, prospectively, for tailored therapies.

MiRNAs may represent a link between the results obtained in animal models and human patients [44]. To explore the hypothesis of miR-146a as biomarkers for ALS and to translate our results to human patients, we first verified whether miR-146a up-regulation in ALS mouse sciatic nerve may be associated with changes in the serum miRNA levels. Of interest, we observed a significant decrease in the miRNA in affected mice compared to controls at the symptomatic disease stage, thus implying an enrichment of miR-146a in the sciatic nerve but not in the circulation. We, thus, investigated miR-146a levels in ALS patients’ serum samples and obtained molecular results indicative of a significant down-regulation of the miRNA compared to healthy controls, in line with the animal model data. Down-regulation of miR-146a was more evident in the serum of ALS patients with UMN predominance compared to LMN patients. By ROC curve analysis, we revealed a potential role of miR-146a as a biomarker for ALS. Indeed, we obtained sensitivity and specificity results indicative of the ability of serum miR-146a levels to discriminate between ALS patients and controls and between patients with UMN or LMN predominance. These findings strongly suggest that miR-146a might be implicated in human ALS pathogenic events and could participate in axon degeneration, as observed in the G93A-SOD1 model.

Serum NFL levels were found to be elevated in ALS patients and to mirror the severity of axonal degeneration, corroborating their value as disease biomarkers [14,15,16]. Similarly, miR-146a might represent a circulating biomarker for the disease, whose levels in serum follow an opposite trend to that of NFL, likely in view of a negative miR-146a/NFL correlation, as observed in our study. Reduced levels of the miRNA in ALS serum could be explained by its enrichment in the peripheral nerve with a lack of release in the blood. However, this issue needs to be deeply explored in a further study, taking into account the complexity of the regulatory mechanisms involving miRNAs and their release in the serum and additional sites in which miR-146a expression could be altered to influence its levels in circulation. Nevertheless, our serum data, and their consistency across the animal model and human patients, strongly support a role for miR-146a as a non-invasive molecular biomarker in ALS, particularly for stratifying ALS patients according to the pattern of MN involvement underlying the clinical phenotype.

The discovery of a possible miR-146a contribution to peripheral axon damage could have relevant therapeutic implications. Considering that MN axon degeneration is an early phenomenon in ALS [5,7] and that miR-146a up-regulation is also an early event, an advanced molecular approach specifically targeting this miRNA could represent an innovative future method to be tested and developed for counteracting the peripheral degenerative cascade of events occurring in the disease. Greater understanding in this research area could have relevant implications for ALS treatment.

## 4. Materials and Methods

### 4.1. Animals

All animal experiments were carried out in accordance with the EU Directive 2010/63 and with Italian law (D.L. 26/2014). Transgenic G93A-SOD1 (B6SJL-Tg (SOD1*G93A)1Gur/J) and control B6.SJL mice were purchased from Charles River Laboratories, Inc. (Wilmington, MA, USA), maintained, and bred at the animal house of the Fondazione IRCCS Istituto Neurologico Carlo Besta. The project was approved by the Ethics Committee of the Institute and the Italian Ministry of Health (ref. 03/2018, 78/2022-PR). Transgenic G93A-SOD1 progenies were identified by quantitative real-time PCR amplification of the mutant human SOD1 gene as previously described [45]. They were sacrificed for tissue collection by exposure to CO_2_ at week 8 (pre-symptomatic stages of disease), week 12 (onset of disease), and week 18 (late stage of disease) [43].

### 4.2. NSC-34 Motor Neuron-like Cell Culture and Differentiation

NSC-34 motor neuron-like cells are hybrid cells produced by the fusion of motor neurons of the spinal cords of mouse embryos with mouse neuroblastoma cells N18TG2 [46]. NSC-34 cells were maintained in DMEM high glucose with 10% fetal bovine serum (FBS) (Thermo Fisher Scientific, Waltham, MA, USA), 1% glutamine, 1% sodium pyruvate, and 1% penicillin-streptomycin (P/S) (Euroclone, Milan, Italy), in a humidified atmosphere and 5% CO_2_ at 37 °C (hereafter referred to as standard culture conditions). NSC-34 was grown for a maximum of 20 passages. To induce NSC-34 differentiation, cells were plated onto Matrigel-coated plates (Corning, Glendale, AZ, USA) in differentiating medium composed of DMEM/F12, 1% FBS (Thermo Fisher Scientific), 1% P/S, 1% sodium pyruvate, 1% MEM non-essential amino acids (Euroclone), supplemented with 1 μM all-trans RA (Sigma-Aldrich, St. Louis, MO, USA) for 12 days, as previously described [47]. The medium was replaced every 2 days.

### 4.3. Transfection of miR-146-5p Inhibitor

A total of 4 × 10^4^ NSC-34 were seeded into Matrigel-coated 6-well plates and differentiated for 12 days. miR-146a-5p mirVana miRNA inhibitor (IDMH10722, Thermo Fisher Scientific) and Silencer Select Negative Control (Thermo Fisher Scientific), as scrambled miRNA, were allowed to form transfection complexes with Lipofectamine RNAiMax transfection reagent (Thermo Fisher Scientific) in Opti-MEM I Reduced Serum Medium (Thermo Fisher Scientific) at a final concentration of 30 nM for 15 min at room temperature. After 24 h, the medium was replaced with NSC-34 differentiation medium; after 48 h, samples from the 6-well plates were collected for RNA extraction. Three replicates per condition were performed.

### 4.4. Quantitative Real-Time PCR to Assess MiR-146a Expression in Mouse Sciatic Nerve Tissue

Total RNA was extracted with Trizol reagent from sciatic nerve tissues (3–4 mg). Sciatic nerve tissues were maintained at −80 °C until use. Total RNA was reverse-transcribed to cDNA using the TaqMan microRNA Reverse Transcription Kit (Thermo Fisher Scientific) with specific primers for miR-146a-5p (termed miR-146a throughout the article). cDNA aliquots corresponding to 15 ng total RNA were amplified by quantitative real-time PCR in duplicate, with Universal PCR Master Mix and specific pre-designed TaqMan microRNA assays (Thermo Fisher Scientific). As endogenous control for data normalization, we used U6 snRNA, which was stably expressed in the G93A-SOD1, and control sciatic nerve tissues. MiR-146a levels were expressed as relative values normalized toward U6, according to the following formula 2^−ΔCt^ × 100.

### 4.5. Fluorescent In Situ Hybridization to Detect MiR-146a in Mouse Sciatic Nerve

Sciatic nerves were collected from G93A-SOD1 and B6.SJL mice at weeks 8, 12, and 18. Sciatic nerves were immediately frozen in isopentane, pre-cooled in liquid nitrogen, and stored at −80 °C. Frozen tissues were then cut into 10 μm thick sections and stored at −80 °C until usage. Sciatic nerve sections were then fixed with paraformaldehyde 4% and then permeabilized with cold methanol (Merck, Darmstadt, Germany). Slides were then washed with Saline-sodium citrate (SSC) buffer, composed of 3 M NaCl and 0.3 M sodium citrate (Carlo Erba Reagents, Milan, Italy) in distilled water and with PBS. Sciatic nerve slices were then covered with miRCURY LNA miRNA Detection Probes 5′-fluorescein and 3′-fluorescein labeled (Qiagen, Hilden, Germany), specific for miR-146a-5p (80 nM, sequence UGAGAACUGAAUUCCAUGGGUU), U6 (20 nM) and scramble (80 nM), resuspended in situ hybridization (ISH) buffer (Qiagen) overnight at 37 °C. U6 was used as endogenous control, and scramble as negative technical control. The slides were then washed with ISH buffer and PBS and then blocked with a BSA solution (5%) (Thermo Fisher Scientific) at room temperature for 1 h. Next, 4,6-diamidino-2-phenylin-dole (DAPI) (1:1000) (Thermo Fisher Scientific) was applied for 5 min to stain cell nuclei. The slices were then mounted with FluorSave (Merck) and air dry overnight. Images were acquired using the C1 laser scanning confocal microscope system (Nikon, Minato City, Tokyo, Japan) and analyzed using Image J software (version 1.8.0_172). Quantification of the immunofluorescence intensity of miR-146a in sciatic nerve tissues was calculated by correcting for background and normalizing to U6 immunofluorescence intensity as endogenous control, using Image J (version 1.8.0_172).

### 4.6. Quantitative Real-Time PCR to Assess Nfl Gene Expression in Mouse Sciatic Nerve Tissue and in NSC-34 Transfected Cells

Total RNA was extracted with Trizol reagent from NSC-34-treated cells (4 × 10^4^ cells per sample). For neurofilament light chain (*Nfl*) gene expression analysis, total RNA from NSC-34-treated cells, and the total RNA previously analyzed for miR-146a from mouse sciatic nerve tissue was retro-transcribed using SuperScript VILO cDNA Synthesis kit (Thermo Fisher Scientific). cDNA (10 ng) was amplified by real-time PCR in duplicate, with TaqMan Fast Advanced Master Mix and TaqMan gene expression assays, specific for *Nfl* on the ViiA7 Real-time PCR system (Thermo Fisher Scientific). The 18S was stably expressed in the G93A-SOD1 and control sciatic nerve tissue and in the NSC-34 cell line and used as endogenous control. Transcriptional levels of the *Nfl* gene were expressed as relative values normalized toward 18S levels, according to the following formula 2^−ΔCt^ × 100.

### 4.7. Western Blotting Assay

To obtain total proteins, sciatic nerves were homogenized in a lysis buffer composed of distilled water supplemented with sodium chloride (NaCl) (150 mM) (Merck), Nonidet P-40 (1%) (Merck), sodium deoxycholate (0.5%) (Merk), sodium dodecyl sulfate (SDS) (0.1%) (Merck), Tris hydrochloride (50 mM, pH 8.0) (Merck), and Halt Protease and Phosphatase Inhibitor Cocktail (100×) (Thermo Fisher Scientific), using Tissue Lyser LT and stainless steel beads (Qiagen) for 3 min at 50 Hz. Extracts were incubated on ice for 30 min and then centrifuged for 20 min at 14,000 rpm at 4 °C to remove particulate matter. Supernatant protein concentration was determined by the Bradford method (Coomassie Plus Assay Kit, Thermo Fisher Scientific ). Western blot analysis was performed on NuPAGE4–12%, Bis-Tris, 1.5 mm, Mini Protein Gels (Thermo Fisher Scientific), loading 30 µg of total proteins, previously denatured at 70 °C for 10 min. Samples were then electrotransferred to iBlot™ 2 Transfer Stacks, PVDF, mini (Thermo Fisher Scientific) using the iBlot 2 Dry Blotting System (Thermo Fisher Scientific). Membranes were treated with a blocking solution containing 5% skim milk powder in Tris-buffered saline with 0.1% Tween 20 (TBS-T) for 1 h at room temperature. Membranes were then incubated with the primary antibodies: anti-β-actin, rabbit polyclonal (dilution 1:3000, ab8227, Abcam, Cambridge, U.K.); anti-NFL, mouse monoclonal dilution (1:1000, MA1-2010, Thermo Fisher Scientific) overnight at 4 °C. Immunoreactivity was detected using secondary Western blot fluorescent antibodies: goat anti-rabbit red (LI-COR Biosciences, NE, USA, dilution 1:10,000) was used to identify β-actin; goat anti-mouse green (LI-COR Biosciences, dilution 1:10,000) was used to identify NFL. Immunoreactive NFL bands were visualized by Odyssey Infrared Imaging System (LI-COR Biosciences) and quantified using the Image Lab software (Bio-Rad Laboratories, Hercules, CA, USA).

### 4.8. Quantitative Real-Time PCR to Assess miR-146a Expression in Mouse Serum

Total RNA was extracted with the miRNeasy Serum/Plasma Kit (Qiagen) from mouse serum. Mouse blood was collected in serum separation tubes, and sera were isolated by incubating the whole blood for 30 min at room temperature and then centrifuging it for 10 min at 3000 rpm at 4 °C. The supernatant was collected as serum. Sera were maintained at –80 °C until use. Total RNA was reverse-transcribed to cDNA using TaqMan microRNA Reverse Transcription Kit (Thermo Fisher Scientific) with specific primers for miR-146a-5p (termed miR-146a throughout the article), and cDNA aliquots were amplified by real-time PCR in duplicate, as described above. As endogenous control for data normalization, we used miR-24, which was stably expressed in G93A-SOD1, and control sera. Levels of miR-146a were expressed as relative values normalized toward miR-24, according to the following formula 2^−ΔCt^.

### 4.9. Patients and Biological Samples

A cohort of 24 clinically defined ALS patients was enrolled in the study, including 9 ALS with UMN predominance; 8 ALS with LMN predominance; 7 ALS with UMN + LMN predominance (OMIM: #105400 Amyotrophic Lateral Sclerosis, ALS1) followed-up at Neurology III Unit, and genetically assessed at Unit of Medical Genetics and Neurogenetics at Fondazione IRCCS Istituto Neurologico Carlo Besta (Milan, Italy). Twenty-three sex- and age-matched healthy controls were included in the analyses. Patient clinical features are reported in Table 1. The study was performed in accordance with the standards of The Code of Ethics of the World Medical Association (Declaration of Helsinki). The investigation was approved by the Ethics Committee of Fondazione IRCCS Istituto Neurologico Carlo Besta (project identification code 92/2019: date January 2019–January 2022). Written informed consent was obtained from each subject. Biological samples were stored at −80 °C in the Biobanks of Fondazione IRCCS Istituto Neurologico Carlo Besta until use.

### 4.10. Quantitative Real-Time PCR to Assess miR-146a Expression in Human Serum

Total RNA was extracted with miRNeasy Serum/Plasma Kit (Qiagen) from human serum. Human blood was collected in serum separation tubes, and sera were isolated by incubating the whole blood for 30 min at room temperature and then centrifuging it for 10 min at 300 rmp at 4 °C. The supernatant was collected as serum. Serums were maintained at −80 °C until use. Expression analysis of miR-146a, and that of miR-24 endogenous control, were performed using the same protocol described for miR-146a analysis in mouse serum, but with primers and TaqMan probe (Thermo Fisher Scientific) specific to humans.

### 4.11. Statistical Analysis

The nonparametric distributed data, verified via Shapiro–Wilk test, were analyzed by Mann–Whitney test for comparison of two groups as indicated in the figure legends. *p*-values < 0.05 were considered statistically significant. A nonparametric Spearman correlation test was applied to evaluate the correlation between expression levels of the miRNA and those of its *Nfl* target gene in the sciatic nerve of G93A-SOD1 mice during disease progression. ROC curves were used to assess the sensitivity and specificity of miR-146a in human serum samples as a biomarker able to discriminate between ALS patients and healthy controls and among UMN, LMN, and UMN + LMN patients. GraphPad Prism version 4.0 (GraphPad Software, San Diego, CA, USA) was used for data elaboration and statistical analysis.

## 5. Conclusions

Our findings suggest that miR-146a may be implicated in early peripheral nerve degeneration in ALS by lowering the levels of NFL. Thus, it could represent a novel target for future molecular strategies for treating the disease or delaying its progression. In addition, we reveal that, along with NFLs; in addition, miR-146a may be a suitable non-invasive biomarker of MN degeneration in ALS. In particular, after confirmation of our data in larger patient cohorts, it could be applied to the clinical practice as a biomarker for patients’ stratification according to UMN or LMN predominance and clinical phenotypes.

## Figures and Tables

**Figure 1 ijms-24-04610-f001:**
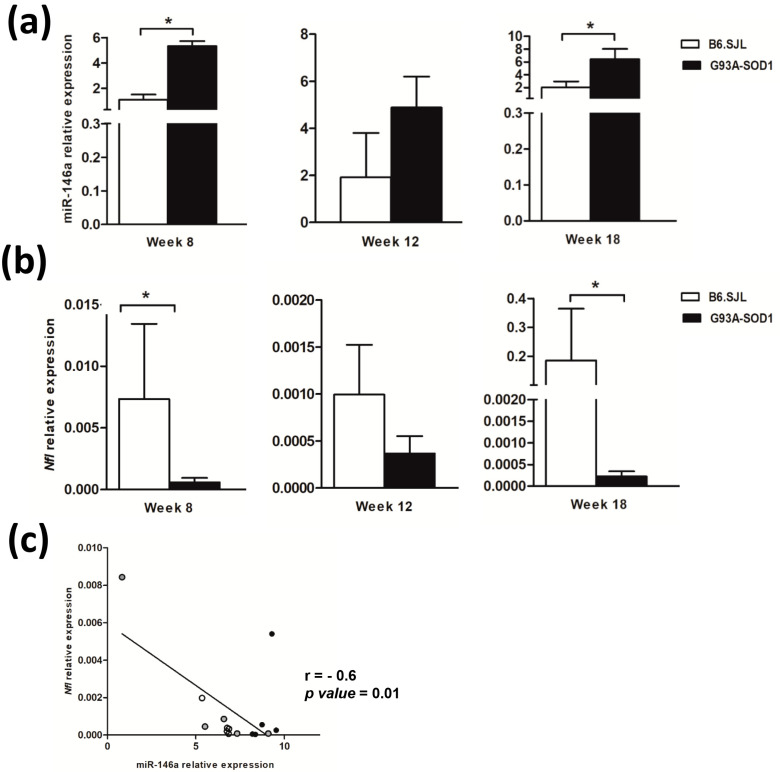
Up−regulation of miR−146a and down−regulation of *Nfl* gene expression in the sciatic nerve of G93A-SOD1 mice. Quantitative real-time PCR analysis of (**a**) miR-146a and (**b**) *Nfl* transcriptional levels in total RNA extracted from the sciatic nerve of G93A-SOD1 (black bars) and B6.SJL mice (white bars) at different stages of the disease, including pre-symptomatic (week 8), onset (week 12), and symptomatic phases (week 18). Five mice per group were included in the analyses. Relative expression data are presented as mean ± SEM of 2^−∆Ct^ × 100 values normalized against the endogenous control U6 snRNA for miR-146a and 18S for the *Nfl* gene. * *p* < 0.05, Mann–Whitney test. (**c**) Significant negative correlation (*p* < 0.05) estimated by Spearman’s correlation test between miR-146a and mRNA levels of *Nfl* in the sciatic nerve of G93A-SOD1 mice at weeks 8 (white circle), 12 (gray circle) and (black circle).

**Figure 2 ijms-24-04610-f002:**
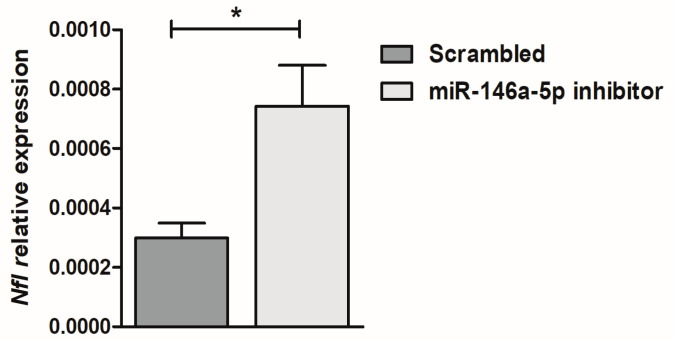
Effect on *Nfl* target transcript of miR-146-5p inhibition in NSC-34 motor neuron-like. Quantitative real-time PCR analysis of *Nfl* transcriptional levels in total RNA extracted from NSC-34 motor neuron-like cells transfected with miR-146a-5p inhibitor (gray bar) and with scrambled miRNA (dark gray bar). Relative expression data are presented as mean ± SEM of 2^−∆Ct^ × 100 values normalized against the endogenous control 18S. * *p* < 0.05, Mann–Whitney test.

**Figure 3 ijms-24-04610-f003:**
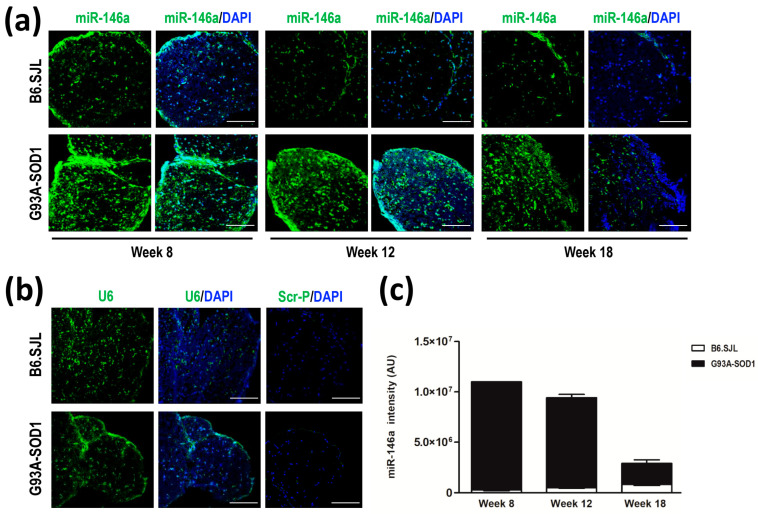
In situ hybridization of miR-146a in the sciatic nerve of G93A-SOD1 and B6.SJL control mice. (**a**) Representative confocal microscopy images of miR-146a (green) in the sciatic nerve of B6.SJL and G93-SOD1 mice at weeks 8, 12, and 18 detected by in situ hybridization (3 mice per group; three independent experiments). Nuclei were revealed by DAPI (blue) counterstaining. (**b**) The left panels show confocal microscopy images of snRNAU6 (green) in the sciatic nerve of B6.SJL and G93A-SOD1 mice (3 mice per group; three independent experiments), analyzed as endogenous control by in situ hybridization; sciatic nerve of animals at weeks 12, as representative for all the time points (week 8 to 18), are shown in the pictures, with nuclei stained by DAPI (blue). The right panels show representative confocal microscopy images of in situ hybridization performed with a scramble probe (green), as a negative control, in the sciatic nerve of B6.SJL and G93-SOD1 mice at week 12, with nuclei stained by DAPI (blue). Scale bars in a and b panels: 20 µm. (**c**) Quantification of the immunofluorescence intensity of miR-146a in G93A-SOD1 (black area) and B6.SJL (white area) sciatic nerve tissues at weeks 8, 12, and 18 (3 mice per group). Density values are reported as mean ± SEM obtained in 3 slides for each animal, corrected for background, and normalized to U6 immunofluorescence intensity.

**Figure 4 ijms-24-04610-f004:**
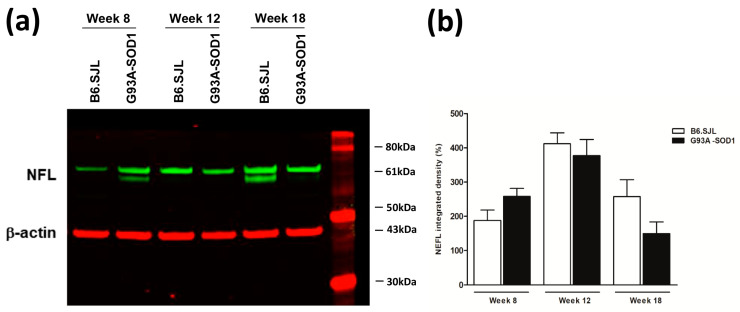
Western blot analysis of NFL protein expression in the sciatic nerve of G93A-SOD1 and B6.SJL control mice. (**a**) Representative Western blot analysis of NFL (green) and β-actin (red) in the sciatic nerve of G93A-SOD1 and B6.SJL control mice at weeks 8, 12, and 18. (**b**) Density values of NFL bands are reported as mean ± SEM (3 mice per group; three independent experiments), corrected for background, and normalized to the β-actin control.

**Figure 5 ijms-24-04610-f005:**
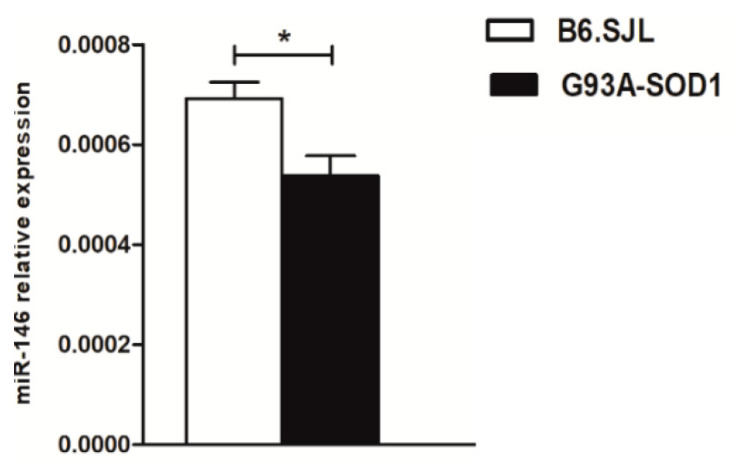
Down−regulation of miR−146a in the serum of ALS mice. Quantitative real-time PCR analysis of miR-146a in total RNA extracted from the serum of G93A-SOD1 (black bars) and B6.SJL mice (white bars) at the symptomatic stage of disease (i.e., week 18; 3 mice per group). Relative expression data are presented as mean ± SEM of 2^−∆Ct^ values normalized against the endogenous control miR-24. * *p* < 0.05, Mann–Whitney test.

**Figure 6 ijms-24-04610-f006:**
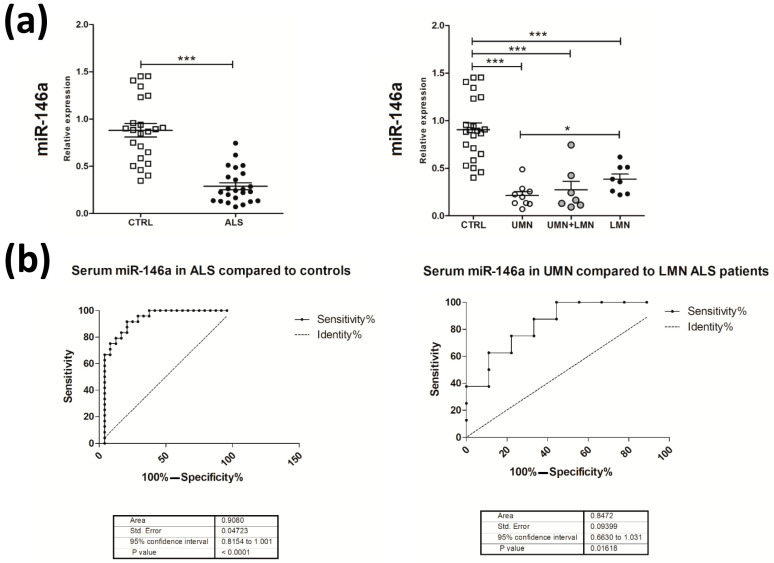
Down-regulation of miR-146a in the serum of ALS patients. (**a**) The **left** panel shows quantitative real-time PCR analysis of miR-146a in total RNA extracted from the serum of ALS patients (black circle) and healthy controls (white squares). The **right** panel shows the same results after stratification of ALS patients according to the involvement of upper (UMN) or lower (LMN) motor neurons in: UMN (white circles), UMN + LMN (gray circles), and LMN (black circles). Relative expression data are presented as mean ± SEM of 2^−∆Ct^ values normalized against the endogenous control miR-24. * *p* < 0.05, *** *p* < 0.001, Mann–Whitney test. (**b**) Receiver operating characteristic (ROC) curves used to assess the sensitivity and specificity of serum miR-146a as a biomarker for ALS (**left** panel) and for discriminating ALS patients with UMN and LMN predominance (**right** panel).

**Table 1 ijms-24-04610-t001:** Main features of the ALS patients participating in the study.

	ALS Patients (*n* = 24)	Healthy Controls (*n* = 23)
**Sex**		
M%	11(45.83%)	11 (47.82%)
**Age** (years, mean ± SEM)	61 ± 8.19	60.17 ± 7.13
**Clinical predominance**		
UMN	9 (37.5%)	
LMN	8 (33.33%)	
UMN + LMN	7 (29.16%)	
**Disease duration**—months (mean ± SEM)		
Whole population	17.93 ± 12.96	
UMN	19.56 ± 12.29	
LMN	21.13 ± 18.24	
UMN + LMN	12.43 ± 9.66	
**ALSFRS-R**		
Whole population	40.71 ± 5.18	
UMN	40.78 ± 6.26	
LMN	42.25 ± 3.11	
UMN + LMN	39.57 ± 3.31	
**Genotype**		
*C9orf72*	1 (4.16%)	
*SOD1*	1 (4.16%)	
*TDP43*	1 (4.16%)	
VUS SOD	2 (8.33%)	
VUS UBQLN2	1 (4.16%)	
No genetic cause is found	18 (75%)	

All data included in the table relate to the time of sampling. M, male; UMN, upper motor neuron; LMN, lower motor neuron; ALSFSR-R, Revised Amyotrophic Lateral Sclerosis Functional Rating Scale; VUS, variant of uncertain (or unknown) significance.

## Data Availability

Data are available from the corresponding author on reasonable request from any qualified investigator. All patients’ data from this study will be shared anonymized in accordance with the consent provided by participants on the use of confidential data. Data reuse is permitted only for academic purposes.

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
