# Peer review of "MiR-146a in ALS: Contribution to Early Peripheral Nerve Degeneration and Relevance as Disease Biomarker"

_ijms, 2023, doi:10.3390/ijms24054610_

Round 1

Reviewer 1 Report (Previous Reviewer 1)

I have read with interest the resubmitted manuscript entitled “MiR-146a in ALS: contribution to early peripheral nerve degeneration and relevance as disease biomarker” by Giagnorio et al. As I have stated before, it is an interesting and potentially important study that could shed further light on some of the roles certain microRNAs play in the axonal degeneration of motoneurons and, consequently, the pathogenesis of ALS. The fact that the study uses both mouse tissues and human samples from healthy and ALS patients certainly strengthens the manuscript.

In my initial criticism I felt to voice the need to include some neurofilament light chain protein immunohistochemistry or NFL gene expression experiments to enhance the impact of the mouse study but that was not how the authors felt. I do think that these experiments could have given the more impact to the manuscript, but I can live without these experiments. To balance this out, however, the authors added a new method and included a new graph to the manuscript on the effect on Nfl target transcript of miR-146-5p inhibition in NSC-34 motor neuron-like cells as a proof that miR-146a-5p negative targeting activity on Nfl gene was not a coincidence observed in ALS mice but a potential direct correlation between miRNA and mRNA. I believe this was a major addition that obviously enhanced the manuscript.

Some minor problems still exist:

Line 129: MiR-146a Increases, and Nfl Gene Expression Decreases…, instead of increase, decrease

Line 451: What are stainless steel “glass” beads? Qiagen sells stainless steel or tungsten/carbide beads but not that steel/glass chimera…

In the present, revised version Table 1 includes the parameter “Age” as the mean ± SD and the parameter “Disease duration” as the mean ± SEM. In the previous version, these parameters were reported as the mean ± SD. Curiously, both the SD and SEM values spread out the same in these two versions of the manuscript (values after the ± signs are the same). Why was SD changed to SEM for the “Disease duration” in the new version?

Author Response

Referee # 1:

I have read with interest the resubmitted manuscript entitled “MiR-146a in ALS: contribution to early peripheral nerve degeneration and relevance as disease biomarker” by Giagnorio et al. As I have stated before, it is an interesting and potentially important study that could shed further light on some of the roles certain microRNAs play in the axonal degeneration of motoneurons and, consequently, the pathogenesis of ALS. The fact that the study uses both mouse tissues and human samples from healthy and ALS patients certainly strengthens the manuscript.

In my initial criticism I felt to voice the need to include some neurofilament light chain protein immunohistochemistry or NFL gene expression experiments to enhance the impact of the mouse study but that was not how the authors felt. I do think that these experiments could have given the more impact to the manuscript, but I can live without these experiments. To balance this out, however, the authors added a new method and included a new graph to the manuscript on the effect on Nfl target transcript of miR-146-5p inhibition in NSC-34 motor neuron-like cells as a proof that miR-146a-5p negative targeting activity on Nfl gene was not a coincidence observed in ALS mice but a potential direct correlation between miRNA and mRNA. I believe this was a major addition that obviously enhanced the manuscript.

Some minor problems still exist:

Line 129: MiR-146a Increases, and Nfl Gene Expression Decreases…, instead of increase, decrease

Line 451: What are stainless steel “glass” beads? Qiagen sells stainless steel or tungsten/carbide beads but not that steel/glass chimera…

In the present, revised version Table 1 includes the parameter “Age” as the mean ± SD and the parameter “Disease duration” as the mean ± SEM. In the previous version, these parameters were reported as the mean ± SD. Curiously, both the SD and SEM values spread out the same in these two versions of the manuscript (values after the ± signs are the same). Why was SD changed to SEM for the “Disease duration” in the new version?

We wish to thank the Reviewer for the comments. We now change the sentences as indicated by the Reviewer.

Reviewer 2 Report (Previous Reviewer 2)

The authors fulfilled the minimum requirements for publication of this study by performing an additional experiment. They show that inhibition of miR-146a-5p indeed induces higher level of Nfl mRNA. However, the statements they introduced in the results and discussion sections are in no way supported by their data. Yes, it seems that there is some kind of link between miR-146a-5p and Nfl mRNA, but this could be due to multiple reasons. Proving that Nfl mRNA is a direct target of miR-146a-5p requires much more work.

The authors clearly have to state (i) that Nfl mRNA is not a predicted target of miR-146a-5p, and (ii) that, nevertheless, they found some kind of relation of Nfl mRNA expression and miR-146a-5p activity that has not been further characterized in this study.

Author Response

Referee # 2:

The authors fulfilled the minimum requirements for publication of this study by performing an additional experiment. They show that inhibition of miR-146a-5p indeed induces higher level of Nfl mRNA. However, the statements they introduced in the results and discussion sections are in no way supported by their data. Yes, it seems that there is some kind of link between miR-146a-5p and Nfl mRNA, but this could be due to multiple reasons. Proving that Nfl mRNA is a direct target of miR-146a-5p requires much more work.

The authors clearly have to state (i) that Nfl mRNA is not a predicted target of miR-146a-5p, and (ii) that, nevertheless, they found some kind of relation of Nfl mRNA expression and miR-146a-5p activity that has not been further characterized in this study.

We are aware that further functional studies are needed to establish that Nfl is a direct target of miR-146-5p and we have added a sentence on this need in the discussion and revised the result section.

This manuscript is a resubmission of an earlier submission. The following is a list of the peer review reports and author responses from that submission.

Round 1

Reviewer 1 Report

The manuscript entitled “MiR-146a in ALS: contribution to early peripheral nerve degeneration and relevance as disease biomarker” by Giagnorio et al. is an interesting and potentially important study that could shed further light on some of the roles certain micro RNAs play in the axonal degeneration of motoneurons and, consequently, the pathogenesis of ALS. It is important to note that, apart from samples of a mouse model of ALS, the study used human samples from healthy and ALS patients as well.

The results, supported by four figures that all consist of several panels, are convincing. The table details the main characteristics of ALS patients. The discussion is thorough, and the references are appropriate.

Perhaps the only shortcoming of the manuscript is that apart from the western blot analysis, neurofilament light chain protein immunohistochemistry or gene expression using in situ hybridization experiments could have been performed to enhance the impact of the mouse study. Based on data seen in Figure 3a, such experiments could also have shown the obvious differences.

Major problem

The numbers of independent/separate experiments are carefully avoided (except for line 160, but here it is not clear whether this applies only to Figure 1b). They should be clearly stated for every type of experiment.

Minor problems

Figure 2a and b should be made larger in order better appreciate the confocal images. I suppose the scale bars are the same in panel a and b…

Line 129: increases, decreases

Line 225: Table 1 should indicate SEM (?)

Line 238: should be (c)

Lines 385, 388: why two versions of ImageJ were used?

Line 398: the section on western blotting does not mention the number of separate experiments, and only one hi-res western blot picture was uploaded… In an ideal case, all western blots images should be uploaded as supporting material.

Line 412: Tris-buffered

Author Response

We wish to thank the Reviewer for the useful comments, which allowed us to improve our manuscript. Please, find below our point-by-point reply to the Reviewers’ comments.

Referee # 1:

The manuscript entitled “MiR-146a in ALS: contribution to early peripheral nerve degeneration and relevance as disease biomarker” by Giagnorio et al. is an interesting and potentially important study that could shed further light on some of the roles certain micro RNAs play in the axonal degeneration of motoneurons and, consequently, the pathogenesis of ALS. It is important to note that, apart from samples of a mouse model of ALS, the study used human samples from healthy and ALS patients as well.

The results, supported by four figures that all consist of several panels, are convincing. The table details the main characteristics of ALS patients. The discussion is thorough, and the references are appropriate.

Perhaps the only shortcoming of the manuscript is that apart from the western blot analysis, neurofilament light chain protein immunohistochemistry or gene expression using in situ hybridization experiments could have been performed to enhance the impact of the mouse study. Based on data seen in Figure 3a, such experiments could also have shown the obvious differences.

We wish to thank the Reviewer for the positive comments on our manuscript.

As regard to neurofilament light chain (NFL) data, gene expression data obtained by real-time PCR shown in Figure 1, and results from Western Blot analysis shown in Figure 3, make us confident on early NFL changes associated with the disease in ALS mice.

Major problem

The numbers of independent/separate experiments are carefully avoided (except for line 160, but here it is not clear whether this applies only to Figure 1b). They should be clearly stated for every type of experiment.

We wish to thank the Reviewer for this useful suggestion. In the revised manuscript, we have now carefully added and indicated in each figure legend the numbers of the independent experiments we performed.    

Minor problems

Figure 2a and b should be made larger in order better appreciate the confocal images. I suppose the scale bars are the same in panel a and b…

We wish to thank the Reviewer for this comment. We have now increased the size of Figure 2a and b. We confirm that the scale bars are the same in all panels (a and b), corresponding to 20 µm, as stated in the figure legend.

Line 129: increases, decreases

Line 225: Table 1 should indicate SEM (?)

Line 238: should be (c)

Lines 385, 388: why two versions of ImageJ were used?

We are sorry for the mistakes. We have now corrected all the sentences; as regard to the line 129, in the 2.1 paragraph title we meant “increment and decrement”, as for the title of the 2.3 paragraph.

Line 398: the section on western blotting does not mention the number of separate experiments, and only one hi-res western blot picture was uploaded… In an ideal case, all western blots images should be uploaded as supporting material.

We have now included Supplemental Figure S1 including original uncropped western blot images obtained from three separate experiments, and specified this number in Figure 3 legend.

Line 412: Tris-buffered

We are sorry for the mistake. We have now corrected the word.

Reviewer 2 Report

In this study, the authors suggest that, in ALS, downregulation of NFL is caused by increased miR-146a, and that miR-146a may represent a novel biomarker for human ALS. Unfortunately, there is a serious flaw in this study preventing interpretation of the results. The authors state that is has been shown that NFL is a target of miR-146a. While this is true for miR-146a-3p (also known as miR-146a*), the authors measured miR-146a-5p that is not related to NFL. Therefore, upregulation of miR-146a and downregulation of NFL mRNA is most likely coincidence.

Additionally, other results in this study need clarification:

- sciatic nerves: the authors state that they used 3-10 mg sciatic nerve tissue for their analyses. This is a lot, other studies report that sciatic nerves are much more lightweight.

- Fig. 3: I do not see on the blot what the authors are stating in the text.

- Fig 4c: an AUC of 0.9 is very good and when looking at Fig. 4b, there's substantial overlap of controls and ALS patients. The authors should carefully re-calculate their results.

Author Response

We wish to thank the Reviewer for the useful comments, which allowed us to improve our manuscript. Please, find below our point-by-point reply to the Reviewers’ comments.

Referee # 2:

In this study, the authors suggest that, in ALS, downregulation of NFL is caused by increased miR-146a, and that miR-146a may represent a novel biomarker for human ALS. Unfortunately, there is a serious flaw in this study preventing interpretation of the results. The authors state that is has been shown that NFL is a target of miR-146a. While this is true for miR-146a-3p (also known as miR-146a*), the authors measured miR-146a-5p that is not related to NFL. Therefore, upregulation of miR-146a and downregulation of NFL mRNA is most likely coincidence.

We agree with the Reviewer comment, and are aware that Nfl has been demonstrated as target of miR-146a*/-3p. However, as we have now discussed in the revised manuscript, 5p/3p types of the same miRNA were found to be co-expressed and co-target the same transcripts. Moreover, interaction between miRNA-5p and -3p with mRNAs can be fully complementary, since binding characteristics of different miRNA-5p/3p pairs in complementary binding sites of the same genes have been established, supporting 5p/3p co-targeting (Huang et al., 2014; Yurikova et al., 2019).  Based on this assumption, decreased expression of Nfl we observed in association with miR-146a-5p increase, could well be associated to the action of this miRNA, as a result of a co-targeting process with miR-146a-3p. Negative correlation between the miRNA and the target gene expression seems to be more than a coincidence, though we cannot exclude NFL changes as a secondary effect, rather than a direct effect, of miR-146a-5p dysregulation in ALS mice.  

Additionally, other results in this study need clarification:

- sciatic nerves: the authors state that they used 3-10 mg sciatic nerve tissue for their analyses. This is a lot, other studies report that sciatic nerves are much more lightweight.

We are sorry for this mistake. In the revised Material and Methods section, we have now clarified the weight of the sciatic nerve tissue (3-4 mg).

- Fig. 3: I do not see on the blot what the authors are stating in the text.

We wish to thank the Reviewer for this helpful comment that allows us to be more accurate in describing the western blot analysis (Figure 3) in the Result section of the revised manuscript (line 192 to 195).

- Fig 4c: an AUC of 0.9 is very good and when looking at Fig. 4b, there's substantial overlap of controls and ALS patients. The authors should carefully re-calculate their results.

We wish to thank the Reviewer for this advice. We have re-calculated the results and confirm AUC value. 

Round 2

Reviewer 2 Report

The authors addressed my comments and the manuscript slightly improved. However, there are still two major issues that prevent publication.

1. The finding that in some special cases the 5p and 3p strand of miRNAs target the same transcript cannot be generalized. Introduction of two more references is surely not sufficient here to indicate that upregulation of miR-146a-5p is related to downregulation of NFL mRNA. Here, additional experiments are mandatory, especially because different algorithms not even predict NFL mRNA as a target of miR-146a-5p. The minimum requirement is to tranfect cells (primary neurons and/or neuronal cell lines) with mimics of miR-146a-5p, and to measure level of NFL mRNA. Without such an experiment, one could choose any deregulated miRNA/mRNA and claim causality. I do not expect the authors to show that NFL is a direct target of miR-146a-5p, but it has to be shown that upregulation of miR-146a-5p causes downregulation of NFL mRNA, directly or indirectly.

2. The authors still claim that Western blot data in Fig. 3 is "in line" with their mRNA data. When looking at the blots and the quantification, there is no change of NFL protein at weeks 8 and 12. At week 18 there may be a non-signifcant trend (the p-value should be shown here) towards decreased NFL protein. This should clearly and honestly be stated in the text. The discrepancy to the mRNA data can easily be explained by the exceptional stability of NFL protein (if I remember right, half-life is about 3 weeks). This should be discussed by the authors.

Author Response

The authors addressed my comments and the manuscript slightly improved. However, there are still two major issues that prevent publication.

  1. The finding that in some special cases the 5p and 3p strand of miRNAs target the same transcript cannot be generalized. Introduction of two more references is surely not sufficient here to indicate that upregulation of miR-146a-5p is related to downregulation of NFL mRNA. Here, additional experiments are mandatory, especially because different algorithms not even predict NFL mRNA as a target of miR-146a-5p. The minimum requirement is to tranfect cells (primary neurons and/or neuronal cell lines) with mimics of miR-146a-5p, and to measure level of NFL mRNA. Without such an experiment, one could choose any deregulated miRNA/mRNA and claim causality. I do not expect the authors to show that NFL is a direct target of miR-146a-5p, but it has to be shown that upregulation of miR-146a-5p causes downregulation of NFL mRNA, directly or indirectly.

We agree with the Reviewer comment that functional studies are needed to demonstrate the effect of miR-146a-5p on Nfl expression, if direct, as reported for miR-146a-3p, or not. We have now described dysregulation of miR-146a and Nfl gene expression as two associated events occurring in sciatic nerve of ALS mice, without emphasizing that Nfl is target of the miRNA, and have revised the text to allow the reader to understand that the exact relationship between miR-146a-5p and NFL needs to be addressed in further studies.

  1. The authors still claim that Western blot data in Fig. 3 is "in line" with their mRNA data. When looking at the blots and the quantification, there is no change of NFL protein at weeks 8 and 12. At week 18 there may be a non-signifcant trend (the p-value should be shown here) towards decreased NFL protein. This should clearly and honestly be stated in the text. The discrepancy to the mRNA data can easily be explained by the exceptional stability of NFL protein (if I remember right, half-life is about 3 weeks). This should be discussed by the authors.

We wish to thank the Reviewer for the useful comment and suggestion. We have now stated that NFL protein levels showed a trend to be decreased at 18 weak (with no significant p value), but not at 8 and 12 weaks. We have now stated that these results are not in line with those of real-time PCR, considering NFL stability as an explanation for the discrepancy between mRNA and protein data, as the Reviewer suggested.

Round 3

Reviewer 2 Report

Given that the authors are not willing to perform essential experiments, I have to recommend rejection of the study. Measuring two, most likely, unrelated RNAs (Nfl mRNA and miR-146-5p) is not sufficient to "discuss" a possible relation.